

# Developing the developmental origins of health and disease (DOHaD) awareness scale to assess an education module for improving dietary behavior among college students

Kadriye Elif İmre[1] and Aslı Akyol[2]

[1] Department of Nutrition and Dietetics, Kastamonu University, Kastamonu, Turkey
[2] Departmant of Nutrition and Dietetics, Hacettepe University, Ankara, Turkey

## ABSTRACT

**Objective:** This study aimed to develop and validate the Developmental Origins of Health and Disease (DOHaD) awareness scale and examine whether having a DOHaD education module may affect dietary behavior in college students.

**Background:** Some studies conducted within the scope of the DOHaD hypothesis show associations between early-life environmental factors, especially maternal health and nutritional status, with the next generation's health and disease status. Despite the increase in elucidating of the underpinning mechanisms of early life determinants and chronic disease risk, there is limited knowledge on how public perceive and understand DOHaD concepts.

**Methods:** The study consisted of three phases: identification of DOHaD awareness scale components, development and validation, interrater reliability of the scale, and a confirmatory study. Two-hundred college students to confirm face validity of the scale, one-hundred for reproducibility and one-hundred for the confirmatory study. Confirmatory study included a pre-intervention period in which baseline parameters (such as anthropometric measurements, HEI-2015 (Healthy Eating Index-2015), physical activity levels, and DOHaD awareness scale scores) were measured at two different time points, and an educational module on DOHaD concepts was provided to the participants. The study was reported in accordance with the Guidelines for Reporting Reliability and Agreement Studies (GRRAS).

**Results:** Content validity ratio, exploratory factor analysis, and Cronbach's alpha values indicated that a reliable and valid instrument to assess the awareness of DOHaD concepts in college students was developed. Despite the lack of a control group, DOHaD concept education was associated with improved anthropometric measurements, healthier eating parameters, increased physical activity levels, and a better understanding of DOHaD concepts.

**Conclusions:** Translating DOHaD concepts into healthier behaviors can support improvements in lifestyle, and the use of the DOHaD awareness scale may serve as a valuable tool to encourage healthy behaviors among college students.

Corresponding authors
Kadriye Elif İmre,
keimre@kastamonu.edu.tr
Aslı Akyol,
asli.akyol@hacettepe.edu.tr

# INTRODUCTION

Non-communicable diseases (NCDs), including cardiovascular diseases, diabetes, stroke, and cancer, are the leading causes of mortality and morbidity in our century (*Wang & Wang, 2020*; *World Health Organization, 2000*). According to WHO data, almost three quarters of all NCD deaths, and 82% of the 16 million people who died prematurely, occur in low- and middle-income countries (*World Health Organization, 2015*). Preventable environmental factors such as smoking, sedentary lifestyle, unbalanced and inadequate nutrition are the major causes of NCDs (*Kelishadi, 2019*). The need for efficient strategies in preventing of these diseases is evident as physiological, socioeconomical and psychological costs related with NCDs, which depend on several factors, become highly detrimental in modern societies (*Oliveira et al., 2023*).

The Developmental Origins of Health and Disease (DOHaD) hypothesis states that early life and prenatal stages of development can permanently influence the postnatal phenotype and thus change susceptibility to NCDs risk during adulthood (*Barker et al., 2002*). As a result of developmental plasticity, disadvantageous environmental stimulants, in particular maternal malnutrition, may trigger negative alterations in fetal metabolism and induce disease related mechanisms (*Gluckman, Hanson & Low, 2019*). Initial evidence derived from the DOHaD hypothesis revealed that maternal undernutrition marked by low birth weight was associated with increased risk of obesity and NCDs in later life (*Wilkins et al., 2021*). Over the last decade, there has been an increasing body of evidence from DOHaD related research, which indicated that the different aspects of fetal nutritional environment impacts upon the development of every baby. As such, both maternal and paternal undernutrition, high fat diets or obesity exerted negative effects on offspring metabolism (*O'Hara, Gembus & Nicholas, 2021*; *Velazquez, Fleming & Watkins, 2019*). Since these issues are prevalent in most countries, there has been an increasing attempt to translate crucial findings from DOHaD research into public awareness (*McKerracher et al., 2019*). However, some studies also indicate that conditioning on intermediates—through methods such as matching, restriction, stratification, or multivariable adjustment—can introduce bias in perinatal research, leading to misleading associations and incorrect biological conclusions (*Kramer et al., 2017*; *Schisterman, Cole & Platt, 2009*). For instance, one study suggests that the strong inverse relationship between birthweight and later blood pressure may stem from insufficient consideration of random error, selective reporting, and inadequate adjustment for confounding factors (*Huxley, Neil & Collins, 2002*). Additionally, analyses of anthropometric data related to small-for-gestational-age (SGA) births can produce conflicting results depending on the statistical methods used (*Kramer et al., 2017*). Since these relationships are often cited as evidence for the fetal origins hypothesis, some studies critically reevaluate similar findings (*Ananth & Schisterman, 2017*; *Elmrayed et al., 2021*; *Paneth, Ahmed & Stein, 1996*).

Improved understanding of the long-term impact of maternal diet may support strategies for NCD prevention. Epigenetics, defined as modifications in gene expression that occur independently of changes in the DNA sequence, plays a vital role in the framework of DOHaD by demonstrating how early environmental factors, including nutrition, stress, and social conditions, can influence long-term health outcomes, thereby emphasizing the necessity for early intervention and heightened awareness (*Gopal, Alenghat & Pammi, 2024*). Research communities focusing on DOHaD principles mentioned the necessity of DOHaD knowledge translation (*Lynch et al., 2022*; *Molinaro et al., 2021*). Supporting individual behavior changes towards a balanced diet may contribute to a healthier lifestyle in subsequent generations. In this context, *Bay et al. (2012)* reported that adolescents showed 30% more understanding of the link between maternal diet during pregnancy and the health of the fetus in adulthood after modules of course work consisted of DOHaD concepts. A further study conducted to examine the impact of The Healthy Start to Life Education for Adolescents Project revealed sustained positive behavior changes in dietary intake and health-related awareness (*Bay et al., 2017*). In a different randomized control trial, intervention students who followed the LifeLab education programme designed with DOHaD concepts, exhibited greater understanding of DOHaD concepts at 12 months but this engagement did not result in behavior change (*Woods-Townsend et al., 2018*). Although there are various personal and environmental factors influencing knowledge acquisition and desirable behavior alterations, elucidating the importance of nutrition education and establishing measurable goals may help to address the gap for setting consistent behavioral changes. World Health Organization (WHO) has indicated diet as a major behavior risk factor that can be optimized through well-understood, cost-effective and feasible interventions (*World Health Organization, 2010*).

Elucidating the present condition of DOHaD knowledge and its possible influence on young individuals is crucially important in order to be able to set these measurable goals and take necessary precaution against the risk of non-communicable diseases. As indicated by *Oyamada et al. (2018)*, defining the baseline awareness and understanding of DOHaD concepts in young individuals is a required approach but to date only few studies examined the awareness levels of DOHaD concepts in young adults. Moreover, the studies examined the awareness levels of DOHaD concepts used basic questionnaires rather than a systematically developed and validated scale. The DOHaD awareness scale is designed to measure the awareness of healthcare professionals in their educational stages and to foster a broader societal consciousness regarding the prevention of unhealthy nutrition and chronic diseases. This dual purpose emphasizes both the immediate educational impact on professionals and the long-term benefits for future generations. Therefore, this study aimed to develop and validate the DOHaD awareness scale, which consists of twenty items that cover important aspects of DOHaD concepts, including epigenetics, social interactions, and life stages from pre-pregnancy to the age of two. Additionally, the scale was developed to measure the knowledge level related to the concepts underpinning the DOHaD hypothesis in a sample of university students and to examine whether an educational module consisting of DOHaD concepts may affect dietary behavior.

## MATERIALS AND METHODS

### Study design

The study was designed in three phases, aimed at determining validity and reliability, and it was reported according to the Guidelines for Reporting Reliability and Agreement Studies (GRRAS) (*Kottner et al., 2011*). Firstly, identification of DOHaD awareness scale dimensions and its components were defined. Secondly, DOHaD awareness scale was developed and validated. Then, a confirmatory study was carried out to ensure validity of the scale. This research was carried out in Kastamonu University Faculty of Health Sciences, first year Nursing Department female undergraduate students between 2018–2020. This study was approved by the Hacettepe University Non-interventional Clinical Researches Ethics Board (reference number GO 18/764-19).

Phase1: Identification of DOHaD awareness scale dimensions and its components

A comprehensive literature review through PubMed, ISI, ScienceDirect, Scopus and Google Scholar was conducted to identify concepts of DOHaD awareness and its components. The following keywords were used: "developmental origins of health and disease", "developmental stage", "childhood growth", "pregnancy", "pre-conception", "transgenerational", "knowledge translation", "non-communicable disease risk", "primary non-communicable disease prevention", "health literacy". Later, reference lists of the studies with these keywords were assessed for additional related research.

Phase 2: Development and validation of the scale

Using the concepts determined at phase 1 and evaluating existing questionnaires a pool of items was generated to measure DOHaD awareness (*Bay et al., 2012*; *Oyamada et al., 2018*; *Woods-Townsend et al., 2018*; *McKerracher et al., 2020*). Then, item pool was reviewed and twenty items most relevant to the concept were selected. Inclusion for validation items was performed by a panel of five experts who had PhD degrees in Nutrition and Dietetics, and the initial questionnaire assessed for qualitative content validity. The experts were asked to comment on the necessity, relevance, clarity and simplicity to calculate content validity ratio (CVR) of each item with Likert-type phrases scoring 4 for "absolutely appropriate", 3 for "appropriate", 2 for "acceptable after edition" and 1 for "absolutely inappropriate". CVR was calculated by the formula: [sum of experts scored 3 and 4/(sum of experts scored 1 and 2/2)] − 1 (*Lawshe, 1975*). After reviewing and editing two items, CVR reached to an acceptable value of 0.80. Content validity and expert panel review led to acceptation of 20 items. The second draft of the scale consisted of 20 Likert-type items. The 5-point Likert-type scale was designed to increase 1 point for each option from none (one point) to severe (five points) and takes values between 20 and 100 points (no items were reverse scored). Thus, the scores 1, 2, 3, 4 and 5 correspond to "not at all", "slightly", "moderately", "very much" and "extremely", respectively.

In order to assess the face validity of the scale, a convenience sample of 250 female nursing students, aged 23 ± 9 years and similar to the target group, was recruited. The sample size was set at 250 participants, following the guideline of having at least ten times the number of observed variables in the scale (*Roscoe, 1975*). Students were randomly chosen and interviewed to independently evaluate each item for clarity and complexity. All

participants provided written informed consent. The data analysis involved exploratory factor analysis (EFA) to assess construct validity. In order to measure whether the scale can be modeled with a factor, the Kaiser-Meyer-Olkin Test (KMO) and Bartlett's test of sphericity were performed.

The scale's reliability was examined by employing internal consistency measures and a test-retest method. Cronbach's alpha was determined to evaluate the internal consistency of the subscales to ensure reproducibility, the questionnaire was re-administered to 150 students aged 22 ± 10 years with a 2-week interval between tests. The average time taken to complete the questionnaire was 10 min. By the end of this phase, a final draft of the questionnaire comprising 20 items was created.

Phase 3: Confirmatory study

A scale was developed to assess awareness and understanding of DOHaD concepts. Furthermore, it was complemented by an educational module aimed at raising individuals' awareness of the relationship between nutrition and DOHaD, as well as promoting the adoption of healthy eating habits. A different confirmatory study was conducted with 100 first year female nursing school students aged 21 ± 12 years to evaluate the impact of the education module. It should be considered that this study was conducted without a control group, which may act as a limiting factor in interpreting the findings regarding the effectiveness of the educational module. To assess consistency of results, the selected samples were different from those studied in the construct validity study. In this study, firstly, 24-h dietary recall, food frequency questionnaire, 24-h physical activity recall and anthropometric measurement records were taken, the Healthy Eating Index (HEI-2015) (*Krebs-Smith et al., 2018*) score was calculated and the DOHaD awareness scale was applied. Despite the lack of a control group, after 4 weeks, an education module consisting of DOHaD concepts was given to participants by the research group.

The education programme consisted of a PowerPoint presentation including the topics of the description of fetal programming, DOHaD components, functions of placenta, importance of a healthy diet and physical activity, and introduction of dietary guidelines of Turkey. Then, problem-based learning, interactive question-answer and discussion techniques were applied to participants. The participants were given printed materials with narrative, examples and pictures containing the topics mentioned in the PowerPoint presentation. After 6 weeks and 6 months, 24-h dietary recall, food frequency questionnaire, 24-h physical activity recall and anthropometric measurement records were taken, the HEI-2015 score was calculated and the DOHaD awareness scale was applied again. Data collection conducted during February 2019 to October 2020. Confirmatory factor analysis was performed using the same parameters and fit indices as phase 2.

## Anthropometric measurements

The participants' body weights were recorded using a portable scale while they wore minimal clothing and no shoes. Height was measured using a wall-mounted stadiometer. Waist circumference was taken at the narrowest point between the iliac crest and the lowest rib, using an inflexible measuring tape. Hip circumference was taken at the highest point of the hip, parallel to the ground, using an inflexible measuring tape. Waist and hip

circumference measurements were repeated three times, and the average values were calculated. All measurements were conducted as outlined in previous studies (*Hawes, 1992*; *Franco-Villoria et al., 2016*). A single healthcare professional conducted all anthropometric measurements. Body mass index (BMI: weight/height$^2$ kg/m$^2$) and the waist-to-hip ratio were calculated and assessed based on the World Health Organization's classification (*World Health Organization, 2010*, *2020*).

### 24-h dietary recall

The 24-h dietary recalls and food consumption frequency assessments were conducted through face-to-face interviews. The types and amounts of all foods and beverages consumed, along with recipes for prepared dishes and cooking methods, were documented and verified. A photographic atlas was utilized to capture the types and portion sizes of food, fluids, and meals (*Rakicioğlu et al., 2012*). The dietary data were analyzed using the BeBIS-7 software program (Nutrition Information Systems Software), which facilitated the calculation of total energy intake, as well as macro and micronutrients (*Nutrition Information Systems, 2004*).

### Healthy eating index (HEI-2015)

The HEI-2005 was employed to evaluate the dietary quality of participants based on 24-h dietary recall data analysis. The HEI-2005 includes components that represent the major food groups: total fruits (worth five points), total vegetables (five points), total grains (five points), dairy products including soy beverages (10 points), and proteins such as meat, poultry, fish, eggs, soy products (excluding beverages), nuts, seeds, and legumes (10 points). Additional components consist of whole fruits (five points); dark green and orange vegetables, including legumes (five points); whole grains (five points); oils (notably non-hydrogenated vegetable oils as well as those in fish, nuts, and seeds) (10 points); saturated fats (10 points); sodium (10 points); and calories from solid fats, alcoholic beverages, and added sugars (20 points). The scores from the HEI-2005 were categorized as "poor" (≤50), "needs improvement" (51–80), and "good" (>80) (*Krebs-Smith et al., 2019*; *Reedy et al., 2018*).

### 24-h physical activity (PA) recall

Participants were asked to record their PA patterns using a diary method that reveals the time spent in various activities (sleep, very light, medium, high, and very high). Energy expenditure was determined according to the activity type by multiplying the activity duration, PA ratio and PA levels. According to PAL values, participants' physical activity status was classified as light (1.40–1.69), moderate (1.70–1.99) and vigorous (2.00–2.40), respectively (*Food and Agriculture Organization of the United Nations, 1985*).

### Statistical analysis

Exploratory factor analysis (EFA) was used to determine the quantity and quality of the scale factors. The Kaiser–Meyer–Olkin (KMO) test was used to measure sample adequacy. Bartlett's globality test and explained total variance were used to evaluate factor analysis. Varimax rotation method was used in EFA. Factor loads were taken into consideration

while deciding whether items should be deleted or kept. Internal consistency of scale items was determined by calculating the Cronbach alpha factor values. Test–retest reliability of the scale was determined using inter-class correlation (ICC) when the ICC > 0.75 was acceptable.

The significance of the difference between nutrition knowledge levels of matched groups before the education and after the study (20–100 points) was evaluated using t-test (paired) with the hypothesis of education increasing the level of awareness.

Data on qualitative variables were indicated using number/ratio charts, the difference between the groups was indicated using three categories of non-parametric test, and the the Marginal Homogeneity test for two dependent variables. Quantitative variables were indicated using mean, median, standard deviation, and top-bottom charts; evaluation of the normal distribution of data was made using the Kolmogorov–Smirnov test. If variables with more than two quantitative groups show normal distribution, repeated measures variance analysis (repeated measures ANOVA) was used; variables countering the normal distribution were applied to Friedman's two-way ANOVA. The Wilcoxon test was applied in dependent variables non-parametric data to evaluate the difference between the test–retest results. Participants' anthropometric measurements were evaluated over time, and the effects of the obtained results were assessed using Partial eta squared values. Partial eta squared is a measure of effect size that indicates the proportion of variance in the dependent variable that is explained by the independent variable.

In this study, generalized estimating equations (GEE) analysis was performed to evaluate the effects of DoHaD awareness scale and HEI-2015 scores on various dependent variables such as body weight, BMI, waist circumference, body fat percentage, and others over three different time points. The analysis used different correlation structures including AR(1), unstructured, exchangeable, and independent. A multivariate regression analysis was conducted to identify the factors influencing DOHaD awareness. All statistical analyses were conducted using SPSS 24.0 (SPSS Inc., Chicago, IL, USA).

## RESULTS

The exploratory factor analysis indicated that 64% of the variance in the DOHaD awareness scale was explained by two factors. Table 1 shows EFA factor loadings and 20 Likert-type items. The KMO value with a lower limit of 0.50 was found to be 0.890, and the Bartlett's test of sphericity was found to be $P < 0.05$. It was determined that the scale is suitable for factor analysis. In the factor analysis conducted, it was determined that 64% of the scale could be explained under two factors. In the scale modeled as two factors, items numbered 1, 2, 3, 4, 5, 6, 7, 9, 10, 18 measure the importance of health and wellness components and 8, 11, 12, 13, 14, 15, 16, 17, 19, 20 numbered items measure the effect of nutrition on fetal programming. The Cronbach's alpha of the scale was found to be 0.953, indicating a classification of 'very reliable'. The scale appears suitable for assessing DOHaD-related awareness in the target population, though additional testing across diverse groups is warranted. Table 2 shows Cronbach's alpha coefficient and ICC of the DOHaD awareness scale and item-total statistics, respectively.

**Table 1 EFA factor loadings.**

| | Avg | SS | EFA factor loadings | | | α if item deleted |
|---|---|---|---|---|---|---|
| | | | Health | Nutrition | % Variance | |
| Q1. How important is it to be healthy? | 4.26 | 0.597 | 0.712 | 0.268 | 0.578 | 0.911 |
| Q2. How important is it to eat healthy? | 4.17 | 0.637 | 0.719 | 0.307 | 0.612 | 0.909 |
| Q3. How important is it to exercise every day? | 3.53 | 0.822 | 0.790 | 0.024 | 0.624 | 0.913 |
| Q4. How much does your current diet affect your future health? | 3.95 | 0.702 | 0.771 | 0.330 | 0.704 | 0.905 |
| Q5. How much does your current exercise level affect your future health? | 3.65 | 0.783 | 0.778 | 0.304 | 0.698 | 0.902 |
| Q6. How important is nutrition during pregnancy? | 4.19 | 0.631 | 0.670 | 0.528 | 0.728 | 0.905 |
| Q7. How much does your social and emotional state affect your health? | 3.81 | 0.849 | 0.555 | 0.456 | 0.516 | 0.911 |
| Q8. How much does being breastfed affect your future health? | 3.79 | 0.844 | 0.328 | 0.715 | 0.618 | 0.936 |
| Q9. How much does a woman's general health and pre-pregnancy well-being affect the fetus during pregnancy? | 3.95 | 0.744 | 0.639 | 0.543 | 0.704 | 0.903 |
| Q10. How much does a man's general health and general condition before pregnancy affect the fetus during his partner's pregnancy? | 3.23 | 1.033 | 0.649 | 0.284 | 0.501 | 0.914 |
| Q11. How much does a pregnant woman's nutrition during pregnancy affect the health of the fetus? | 3.90 | 0.785 | 0.359 | 0.770 | 0.722 | 0.934 |
| Q12. How much does a pregnant woman's nutrition during pregnancy affect the health of the fetus in childhood? | 3.48 | 0.893 | 0.352 | 0.842 | 0.833 | 0.928 |
| Q13. How much does a pregnant woman's nutrition during pregnancy affect the health of the fetus in adulthood? | 3.16 | 1.042 | 0.240 | 0.793 | 0.686 | 0.933 |
| Q14. How much does a pregnant woman's nutrition during pregnancy affect the health of the fetus in elderly? | 2.95 | 1.058 | 0.184 | 0.727 | 0.562 | 0.939 |
| Q15. How much does a child's nutrition up to the age of two affect the child's health later in life? | 3.65 | 0.796 | 0.300 | 0.757 | 0.663 | 0.935 |
| Q16. How much does a child's nutrition up to the age of two affect the child's health in adulthood? | 3.30 | 0.916 | 0.195 | 0.817 | 0.706 | 0.932 |
| Q17. How much does a person's diet affect the risk of developing non-communicable diseases such as cancer, heart diseases, type 2 diabetes *etc.*? | 3.83 | 0.900 | 0.390 | 0.706 | 0.650 | 0.936 |
| Q18. How important are the first 1,000 days of life in the programming of health and disease? | 3.67 | 0.697 | 0.546 | 0.416 | 0.471 | 0.912 |
| Q19. To what extent does inadequate and unbalanced nutrition of a pregnant woman cause permanent changes in the metabolism of fetus? | 3.69 | 0.873 | 0.354 | 0.823 | 0.802 | 0.930 |
| Q20. How much can a pregnant woman's nutritional status during pregnancy affect the health of the next generations by changing gene functioning? | 3.22 | 0.949 | 0.237 | 0.607 | 0.424 | 0.943 |
| **Eigenvalue** | | | 10.978 | 1.824 | – | – |
| **Explained** | | | 35.785 | 28.227 | – | – |
| **Cronbach's α total** | | | 0.917 | 0.941 | – | – |

As shown in Table 3, the majority of the anthropometric measurements changed after the intervention period. The values of body weight, BMI, waist circumference, hip circumference, mid upper arm circumference, waist/height ratio, body fat mass and total body water measured 6 months after the intervention were significantly lower than the pre-intervention values ($P < 0.001$). These anthropometric parameters did not indicate a significant difference when the values of 6 weeks after intervention compared to the

**Table 2 Cronbach's alpha coefficient and ICC of the DOHaD Awareness scale.**

|  | Number of items | Cronbach's α | ICCa (95% CI) |
| --- | --- | --- | --- |
| Health | 10 | 0.917 | 0.914 [0.892–0.936] |
| Nutrition | 10 | 0.941 | 0.956 [0.962–0.984] |
| DOHaD awareness scale (total) | 20 | 0.953 | 0.949 [0.932–0.993] |

**Note:**
Questions 1, 2, 3, 4, 5, 6, 7, 9, 10 and 18 placed in factor "health" and questions 8, 11, 12, 13, 14, 15, 16, 17, 19 and 20 placed in factor "nutrition".

**Table 3 Anthropometric measurement of participants before, 6 weeks after and 6 months after intervention.**

|  | Pre-intervention | | 6 weeks after intervention | | 6 months after intervention | | P value | Partial eta squared |
| --- | --- | --- | --- | --- | --- | --- | --- | --- |
|  | SD | Median (min–max) | SD | Median (min–max) | SD | Median (min–max) |  |  |
| Body weight (kg) | 56.2[a] | 53.5 (40.1–83.7) | 56.2[a] | 53.3 (39.0–83.8) | 53.9[b] | 53.0 (38.8–78.1) | <0.01* | 0.367 |
| BMI (kg/m$^2$) | 21.8[a] | 21.2 (16.6–32.5) | 21.8[a] | 21.2 (16.8–32.7) | 20.9[b] | 20.7 (16.0–30.5) | <0.01# | 0.366 |
| Waist circumference (cm) | 72.0[a] | 70.0 (61.0–98.0) | 71.9[a] | 71.0 (60.0–96.0) | 70.22[b] | 70.0 (58.0–88.0) | <0.01* | 0.142 |
| Hip circumference (cm) | 95.4[a] | 95.0 (82.0–116.0) | 95.4[a] | 95.0 (82.0–116.0) | 93.4[b] | 93.0 (79.0–115.0) | <0.01# | 0.291 |
| Mid upper arm circumference (cm) | 25.7[a] | 25.0 (20.0–35.0) | 25.7[a] | 25.0 (21.0–35.0) | 24.9[b] | 25.0 (20.0–31.0) | <0.01* | 0.123 |
| Waist/hip ratio | 0.8 | 0.7 (0.7–0.9) | 0.8 | 0.8 (0.7–0.9) | 0.8 | 0.8 (0.7–0.9) | 0.494* | 0.003 |
| Waist/height ratio | 0.5[a] | 0.4 (0.4–0.6) | 0.5[a] | 0.4 (0.4–6.0) | 0.4[b] | 0.4 (0.4–0.5) | <0.01* | 0.148 |
| Body fat (%) | 25.8[a] | 25.3 (11.6–40.0) | 25.6[a] | 25.3 (11.4–39.4) | 24.2[b] | 24.1 (10.3–35.8) | <0.01# | 0.272 |
| Total body water (%) | 53.9[a] | 54.2 (43.6–66.0) | 53.9[a] | 54.1 (43.8–65.7) | 54.7[b] | 55.4 (43.6–68.6) | <0.01# | 0.182 |
| HEI |  |  |  |  |  |  |  | 0.475 |

**Notes:**
[a] Mean values with unlike superscript letters were significantly different ($P < 0.05$).
[b] Mean values with unlike superscript letters were significantly different ($P < 0.05$).
Repeated measures ANOVA (*) and Friedman test (#) indicated significantly different scores of DOHaD Awareness Scale at pre-intervention, 6 weeks after intervention and 6 months after intervention.
The partial eta squared values represent the effect size of the differences observed, indicating the proportion of variance in the dependent variable that can be attributed to the independent variable.

pre-intervention values ($P > 0.05$). According to the partial eta squared values, the findings of the study demonstrate significant changes over time in measurements such as body weight and BMI, with substantial effects attributed to the independent variable (0.367 for body weight and 0.366 for BMI). Other measurements also exhibited similar effectiveness, with the effects of changes in waist circumference and body fat identified as 0.291 and 0.272, respectively. The interventions were associated with notable changes in participants' anthropometric values, although further studies are needed to confirm these effects.

Figure 1 shows the comparison of component scores of HEI-2015 recorded during the pre-intervention, 6 weeks after intervention and 6 months after intervention. Accordingly, participants consumed significantly more total fruit ($P < 0.001$) and whole fruit ($P < 0.001$) after the intervention period. Although scores were higher at 6 months after intervention, the difference between the scores of 6 weeks and 6 months after intervention did not reach to a statistically significant difference. Similarly, scores of total vegetables significantly increased both in 6 weeks ($P < 0.001$) and 6 months ($P < 0.001$) after intervention period.

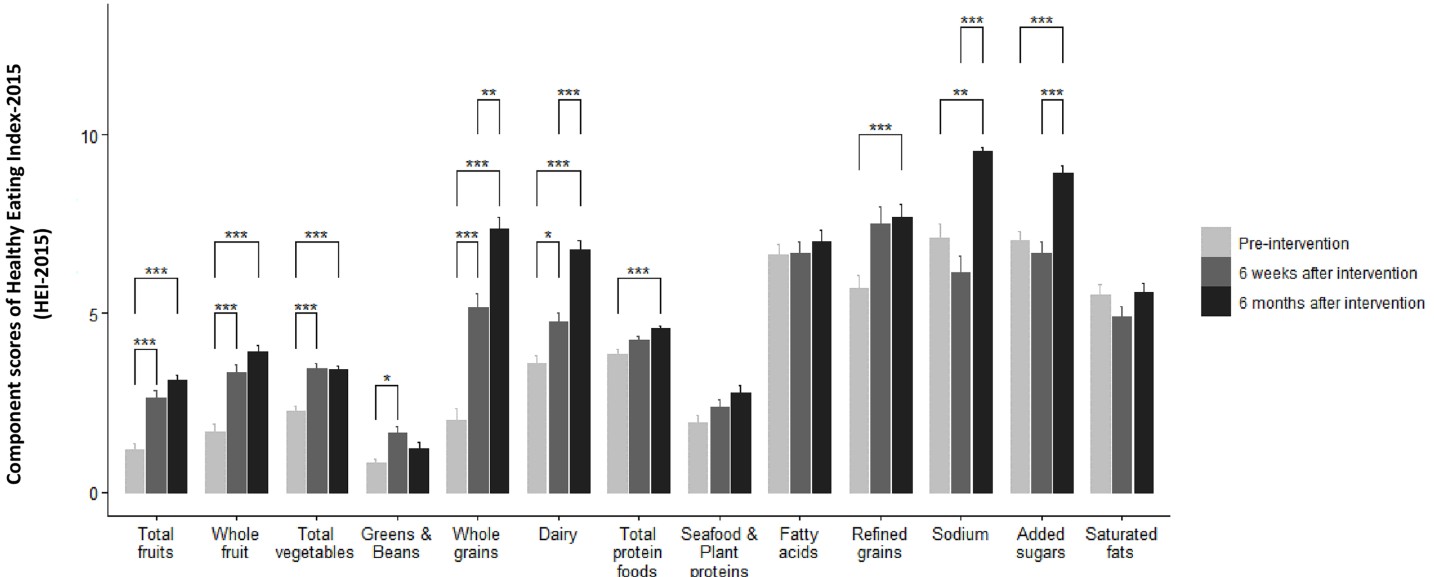

**Figure 1 Comparison of component scores of HEI-2015 measured during pre-intervention, 6 weeks after intervention and 6 months after intervention.** In HEI-2015, a higher score indicates a higher diet quality. The Friedman test indicated significant differences in the component scores of HEI-2015 between pre-intervention, 6 weeks after intervention and 6 months after intervention values. *$P < 0.05$, **$P < 0.01$, ***$P < 0.001$.

The consumption of greens & beans only increased during the period of 6 weeks after intervention significantly ($P < 0.05$) and then decreased at the end of the 6 months after intervention phase. The strongest impact of the intervention was observed on the scores of whole grains and dairy since the consumption of these groups continued to increase significantly both in 6 weeks and 6 months after intervention phase. Total protein foods consumed more only during the 6 months after intervention period ($P < 0.001$). Despite these significant alterations in the component scores of HEI-2015, the intervention did not influence the scores of seafood & plant proteins, fatty acids and saturated fats ($P > 0.05$). Refined grains were the only increased component of HEI-2015 at both phases in the current study. Although the scores of sodium and added sugars seemed to be reduced during the 6 weeks after intervention, they significantly increased at the end of the study period ($P < 0.001$).

Figure 2 demonstrates the average physical activity duration (min/day) of the participants. In addition, moderate exercise duration significantly increased both at 6 weeks ($P < 0.05$) and 6 months ($P < 0.05$) after intervention. On the other hand, duration of vigorous exercise did not exhibit a significant change in all phases of the study ($P > 0.05$).

The effect of the educational intervention on the participants' DoHaD awareness scale scores has been investigated, and the results are presented in Table 4. The scores of the questions that are related to the concepts of DOHaD mostly (Q5. "How much does your current exercise level affect your future health?"; Q6. "How important is nutrition during pregnancy?"; Q8. "How much does being breastfed affect your future health?"; Q9. "How much does a woman's general health and pre-pregnancy well-being affect the fetus during pregnancy?"; Q10. "How much does a man's general health and general condition before

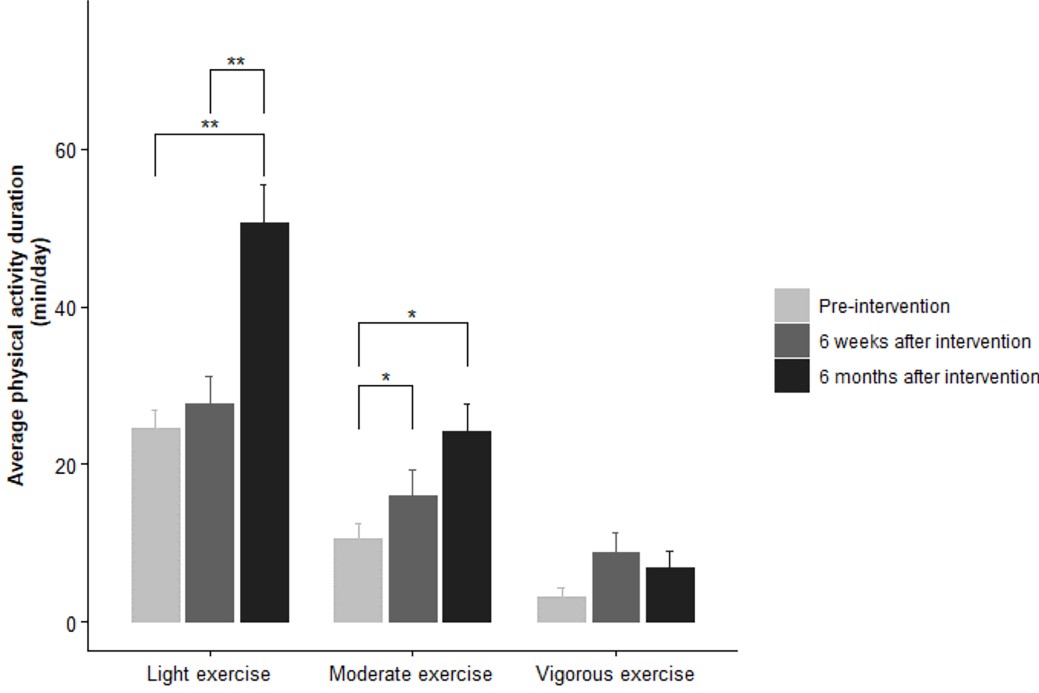

**Figure 2 Comparison of physical activity duration of participants recorded at pre-intervention, 6 weeks after intervention and 6 months after intervention.** The Friedman test indicated significant differences in the average physical activity duration (min/day) between pre-intervention, 6 weeks after intervention and 6 months after intervention values. $*P < 0.05$, $**P < 0.01$.

**Table 4 Comparison of DOHaD Awereness Scale Scores measured at pre-intervention, 6 weeks after intervention and 6 months after intervention.**

|  | Pre-intervention | | 6 weeks after intervention | | 6 months after intervention | | P value |
|---|---|---|---|---|---|---|---|
|  | x̄ ± SD | Median (min–max) | x̄ ± SD | Median (min–max) | x̄ ± SD | Median (min–max) |  |
| Q1. How important is it to be healthy? | 4.3 ± 0.6 | 5 (4–5) | 4.9 ± 0.3 | 5 (4–5) | 4.9 ± 0.2 | 5 (4–5) | >0.05 |
| Q2. How important is it to eat healthy? | 4.7 ± 0.6 | 5 (3–5) | 4.9 ± 0.3 | 5 (4–5) | 4.9 ± 0.3 | 5 (4–5) | >0.05 |
| Q3. How important is it to exercise every day? | 4.5 ± 0.8 | 5 (3–5) | 4.5 ± 0.6 | 5 (3–5) | 4.6 ± 0.6 | 5 (3–5) | >0.05 |
| Q4. How much does your current diet affect your future health? | 4.9 ± 0.7 | 5 (4–5) | 4.8 ± 0.4 | 5 (3–5) | 4.8 ± 0.4 | 5 (4–5) | >0.05 |
| Q5. How much does your current exercise level affect your future health? | 3.7 ± 0.8[a] | 4 (2–5) | 4.6 ± 0.6[b] | 5 (3–5) | 4.7 ± 0.5[b] | 5 (3–5) | <0.001 |
| Q6. How important is nutrition during pregnancy? | 4.2 ± 0.6[a] | 4 (3–5) | 4.9 ± 0.3[b] | 5 (4–5) | 4.9 ± 0.2[b] | 5 (4–5) | <0.001 |
| Q7. How much does your social and emotional state affect your health? | 4.8 ± 0.9 | 5 (2–5 | 4.7 ± 0.6 | 5 (2–5) | 4.7 ± 0.5 | 5 (3–5) | >0.05 |
| Q8. How much does being breastfed affect your future health? | 3.8 ± 0.8[a] | 4 (1–5) | 4.7 ± 0.5[b] | 5 (3–5) | 4.8 ± 0.4[b] | 5 (3–5) | <0.001 |
| Q9. How much does a woman's general health and pre-pregnancy well-being affect the fetus during pregnancy? | 3.9 ± 0.7[a] | 4 (2–5) | 4.8 ± 0.4[b] | 5 (3–5) | 4.9 ± 0.3[b] | 5 (4–5) | <0.001 |

(Continued)

| | Pre-intervention | | 6 weeks after intervention | | 6 months after intervention | | P value |
|---|---|---|---|---|---|---|---|
| | x̄ ± SD | Median (min–max) | x̄ ± SD | Median (min–max) | x̄ ± SD | Median (min–max) | |
| Q10. How much does a man's general health and general condition before pregnancy affect the fetus during his partner's pregnancy? | 3.2 ± 1.0[a] | 3 (1–5) | 4.7 ± 0.5[b] | 5 (3–5) | 4.7 ± 0.6[b] | 5 (3–5) | <0.001 |
| Q11. How much does a pregnant woman's nutrition during pregnancy affect the health of the fetus? | 3.9 ± 0.8[a] | 4 (2–5) | 4.7 ± 0.5[b] | 5 (3–5) | 4.8 ± 0.4[b] | 5 (4–5) | <0.001 |
| Q12. How much does a pregnant woman's nutrition during pregnancy affect the health of the fetus in childhood? | 3.8 ± 0.9[a] | 3 (2–5) | 4.5 ± 0.6[b] | 5 (3–5) | 4.7 ± 0.6[b] | 5 (3–5) | <0.001 |
| Q13.How much does a pregnant woman's nutrition during pregnancy affect the health of the fetus in adulthood? | 3.2 ± 1.0[a] | 3 (1–5) | 4.5 ± 0.6[b] | 5 (3–5) | 4.6 ± 0.6[b] | 5 (3–5) | <0.001 |
| Q14. How much does a pregnant woman's nutrition during pregnancy affect the health of the fetus in elderly? | 2.9 ± 1.1[a] | 3 (1–5) | 4.3 ± 0.7[b] | 4 (2–5) | 4.7 ± 0.5[b] | 5 (3–5) | <0.001 |
| Q15. How much does a child's nutrition up to the age of two affect the child's health later in life? | 3.7 ± 0.8[a] | 4 (1–5) | 4.5 ± 0.6[b] | 5 (3–5) | 4.7 ± 0.5[b] | 5 (3–5) | <0.001 |
| Q16. How much does a child's nutrition up to the age of two affect the child's health in adulthood? | 3.3 ± 0.9[a] | 3 (1–5) | 4.4 ± 0.6[b] | 4 (2–5) | 4.7 ± 0.5[b] | 5 (3–5) | <0.001 |
| Q17. How much does a person's diet affect the risk of developing non-communicable diseases such as cancer, heart diseases, type 2 diabetes etc.? | 3.8 ± 0.9[a] | 4 (1–5) | 4.6 ± 0.6[b] | 5 (3–5) | 4.8 ± 0.5[b] | 5 (3–5) | <0.001 |
| Q18. How important are the first 1,000 days of life in the programming of health and disease? | 3.7 ± 0.7[a] | 4 (2–5) | 4.6 ± 0.6[b] | 5 (3–5) | 4.7 ± 0.5 [b] | 5 (3–5) | <0.001 |
| Q19. To what extent does inadequate and unbalanced nutrition of a pregnant woman cause permanent changes in the metabolism of fetus? | 3.7 ± 0.9[a] | 4 (2–5) | 4.6 ± 0.6[b] | 5 (2–5) | 4.6 ± 0.6[b] | 5 (3–5) | <0.001 |
| Q20. How much can a pregnant woman's nutritional status during pregnancy affect the health of the next generations by changing gene functioning? | 3.2 ± 0.9[a] | 3 (1–5) | 4.5 ± 0.7[b] | 5 (2–5) | 4.5 ± 0.7[b] | 5 (3–5) | <0.001 |

**Notes:**
[a] Mean values with unlike superscript letters were significantly different ($P < 0.05$).
[b] Mean values with unlike superscript letters were significantly different ($P < 0.05$).
Friedman test indicated significantly different scores of DOHaD Awareness Scale at pre-intervention, 6 weeks after intervention and 6 months after intervention.

pregnancy affect the fetus during his partner's pregnancy?"; Q11. "How much does a pregnant woman's nutrition during pregnancy affect the health of the fetus?"; Q12. "How much does a pregnant woman's nutrition during pregnancy affect the health of the fetus in childhood?"; Q13. "How much does a pregnant woman's nutrition during pregnancy affect the health of the fetus in adulthood?"; Q14. "How much does a pregnant woman's nutrition during pregnancy affect the health of the fetus in elderly?"; Q15. "How much does a child's nutrition up to the age of two affect the child's health later in life?"; Q16. "How much does a child's nutrition up to the age of two affect the child's health in adulthood?"; Q17. "How much does a person's diet affect the risk of developing non-communicable diseases such as cancer, heart diseases, type 2 diabetes etc.?"; Q18. "How important are the first 1,000 days of life in the programming of health and disease?"; Q19. "To what extent does inadequate and unbalanced nutrition of a pregnant woman

cause permanent changes in the metabolism of the fetus?"; Q20. "How much can a pregnant woman's nutritional status during pregnancy affect the health of the next generations by changing gene functioning?") significantly increased at 6 weeks ($P < 0.001$) and 6 months ($P < 0.001$) after the intervention. The scores obtained at 6 weeks and 6 months after the intervention did not differ significantly.

The results obtained from the GEE analysis indicated that the DoHaD awareness scale and HEI-2015 scores, along with their interaction, were significant predictors for all dependent variables ($P < 0.001$), and the AR(1) correlation structure exhibited the best model fit for most variables, as evidenced by the lowest QIC values. However, since the findings were evaluated without a control group, it is not possible to determine the effect of the educational module on the results independently of other external factors. The results of the GEE analysis are presented in Table 5. The multivariate regression analysis presented in Table 6 indicates that the waist-to-height ratio is a significant variable influencing DOHaD awareness ($F = 3.718$, $P = 0.038$), while no other anthropometric variables or HEI scores were significantly associated with DOHaD awareness. Additionally, the time variable did not exhibit a statistically significant effect ($F = 0.438$, $P = 0.594$), indicating that time does not have a notable impact on DOHaD awareness.

## DISCUSSION

This article examines the components of the DOHaD awareness scale and investigates its development and validity as a reliable tool for assessing awareness of DOHaD concepts among university students and furthermore, it explores whether participation in an educational module related to these concepts influences dietary behaviors, physical activity levels, and the overall understanding of DOHaD among this demographic. As a result of the study, a reliable and valid instrument has been developed to assess awareness of DOHaD concepts among university students. Our results indicated a positive association between a single DOHaD education module and short-term improvements in anthropometric measurements, healthier eating parameters, light to moderate physical activity levels, and understanding of DOHaD concepts among participants. These parameters were evaluated at two different time points during the study period. These were 6 weeks and 6 months after the educational module. The results suggested initial improvements in anthropometric measurements, HEI-2015 scores, and physical activity levels, with some metrics maintained or modestly enhanced 6 months after the education module. The educational intervention was shown to increase only light to moderate physical activity. The lack of change in intense exercise may be attributed to barriers related to perceived difficulty, individual fitness levels, and external factors. It is advisable for future research and interventions to address these issues. These outcomes indicated that DOHaD concepts were associated with improved health behaviors in college students, with some of these behaviors observed to persist over a 6-month period. This study contributes to understanding DOHaD awareness levels among university students and highlights areas for further investigation. By developing the DOHaD awareness scale, it provides a validated tool for assessment in this demographic. The study suggests

**Table 5 GEE analysis results: model fit indices and correlation structures.** The GEE analysis applies various correlation structures: AR(1) indicates autoregressive correlation where each measurement correlates with the previous one, Unstructured implies no specific correlation assumptions, Exchangeable assumes equal correlation among all measurements, and Independent indicates no correlation between measurements. The QIC (Model Fit Index) assesses the goodness of fit for the model, with lower values suggesting a better fit, while the *P*-value indicates the statistical significance of the model, with values less than 0.05 considered significant.

| Dependent variable | Independent variables | Correlation structure | QIC (model fit index) | *P*-value (model) |
|---|---|---|---|---|
| Body weight | DOHaD Awareness Score, HEİ-2015 Score, DOHaD Awareness*Score, HEİ-2015 Score | AR(1) | 23,081.147 | <0.001 |
| | | Unstructured | 23,211.260 | <0.001 |
| | | Exchangeable | 22,956.384 | <0.001 |
| | | Independent | 22,749.601 | <0.001 |
| BMI | DOHaD Awareness Score, HEİ-2015 Score, DOHaD Awareness*Score, HEİ-2015 Score | AR(1) | 3,146.578 | <0.001 |
| | | Unstructured | 3,150.158 | <0.001 |
| | | Exchangeable | 3,125.798 | <0.001 |
| | | Independent | 3,093.997 | <0.001 |
| Waist circumference | DOHaD Awareness Score, HEİ-2015 Score, DOHaD Awareness*Score, HEİ-2015 Score | AR(1) | 13,444.017 | <0.001 |
| | | Unstructured | 13,519.222 | <0.001 |
| | | Exchangeable | 13,409.018 | <0.001 |
| | | Independent | 13,345.072 | <0.001 |
| Waist height ratio | DOHaD Awareness Score, HEİ-2015 Score, DOHaD Awareness*Score, HEİ-2015 Score | AR(1) | 7.225 | <0.001 |
| | | Unstructured | 7.663 | <0.001 |
| | | Exchangeable | 7.345 | <0.001 |
| | | Independent | 11.024 | <0.001 |
| Hip circumference | DOHaD Awareness Score, HEİ-2015 Score, DOHaD Awareness*Score, HEİ-2015 Score | AR(1) | 13,499.730 | <0.001 |
| | | Unstructured | 134,543.286 | <0.001 |
| | | Exchangeable | 13,454.178 | <0.001 |
| | | Independent | 13,244.445 | <0.001 |
| Waist hip ratio | DOHaD Awareness Score, HEİ-2015 Score, DOHaD Awareness*Score, HEİ-2015 Score | AR(1) | 8.614 | <0.001 |
| | | Unstructured | 8.496 | <0.001 |
| | | Exchangeable | 8.498 | <0.001 |
| | | Independent | 10.734 | <0.001 |
| Body fat % | DOHaD Awareness Score, HEİ-2015 Score, DOHaD Awareness*Score, HEİ-2015 Score | AR(1) | 9,964.448 | <0.001 |
| | | Unstructured | 10,047.985 | <0.001 |
| | | Exchangeable | 9,913.703 | <0.001 |
| | | Independent | 9,764.007 | <0.001 |
| TBW | DOHaD Awareness Score, HEİ-2015 Score, DOHaD Awareness*Score, HEİ-2015 Score | AR(1) | 5,498.383 | <0.001 |
| | | Unstructured | 5,493.678 | <0.001 |
| | | Exchangeable | 5,493.238 | <0.001 |
| | | Independent | 5,421.945 | <0.001 |
| Upper mid-arm circumference | DOHaD Awareness Score, HEİ-2015 Score, DOHaD Awareness*Score, HEİ-2015 Score | AR(1) | 2,291.364 | <0.001 |
| | | Unstructured | 2,305.663 | <0.001 |
| | | Exchangeable | 2,278.884 | <0.001 |
| | | Independent | 2,273.064 | <0.001 |

**Table 6 The effect of time, anthropometric measurements, and HEI-2015 score on DOHaD awareness scale scores.** The table presents the results of the multiple regression analysis, detailing the Type III sums of squares for each source, along with their degrees of freedom (df), mean squares, *F*-statistics, and significance (*P*-values). These results indicate the contribution of each predictor variable to the overall model and their significance in explaining the variance in the dependent variable. The correction method applied is the Greenhouse-Geisser correction, which adjusts the degrees of freedom for tests involving repeated measures to compensate for potential violations of sphericity, thereby providing more accurate F-statistics and significance levels for the associated effects in the multiple regression analysis.

| Source | Type III Sum of squares | df | Mean square | F | Sig. | Correction |
|---|---|---|---|---|---|---|
| Time | 38.329 | 1.535 | 24.943 | 0.438 | 0.594 | Greenhouse-Geisser |
| Body Weight1 | 62.326 | 1.535 | 40.590 | 0.713 | 0.457 | Greenhouse-Geisser |
| Body Weight2 | 181.570 | 1.535 | 118.249 | 2.078 | 0.142 | Greenhouse-Geisser |
| Body Weight3 | 100.097 | 1.535 | 65.189 | 1.145 | 0.311 | Greenhouse-Geisser |
| BMI1 | 51.854 | 1.535 | 33.770 | 0.593 | 0.511 | Greenhouse-Geisser |
| BMI2 | 196.605 | 1.535 | 128.040 | 2.250 | 0.123 | Greenhouse-Geisser |
| BMI3 | 137.734 | 1.535 | 89.700 | 1.576 | 0.215 | Greenhouse-Geisser |
| Waist Circumference 1 | 7.413 | 1.535 | 4.828 | 0.085 | 0.871 | Greenhouse-Geisser |
| Waist Circumference 2 | 5.321 | 1.535 | 3.465 | 0.061 | 0.899 | Greenhouse-Geisser |
| Waist Circumference 3 | 181.075 | 1.535 | 117.926 | 2.072 | 0.142 | Greenhouse-Geisser |
| Hip Circumference 1 | 28.517 | 1.535 | 18.572 | 0.326 | 0.664 | Greenhouse-Geisser |
| Hip Circumference 2 | 8.211 | 1.535 | 5.347 | 0.094 | 0.861 | Greenhouse-Geisser |
| Hip Circumference 3 | 27.979 | 1.535 | 18.221 | 0.320 | 0.669 | Greenhouse-Geisser |
| Waist/Height Ratio1 | 20.641 | 1.535 | 13.443 | 0.236 | 0.730 | Greenhouse-Geisser |
| Waist/Height Ratio2 | 35.379 | 1.535 | 23.041 | 0.405 | 0.614 | Greenhouse-Geisser |
| Waist/Height Ratio3 | 324.957 | 1.535 | 211.630 | 3.718 | 0.038 | Greenhouse-Geisser |
| Waist/Hip Ratio1 | 20.195 | 1.535 | 13.453 | 0.236 | 0.730 | Greenhouse-Geisser |
| Waist/Hip Ratio2 | 40.482 | 1.535 | 13.152 | 0.231 | 0.734 | Greenhouse-Geisser |
| Waist/Hip Ratio3 | 120.424 | 1.535 | 26.364 | 0.463 | 0.579 | Greenhouse-Geisser |
| TBW1 | 50.944 | 1.535 | 33.178 | 0.583 | 0.517 | Greenhouse-Geisser |
| TBW2 | 120.194 | 1.535 | 78.277 | 1.375 | 0.255 | Greenhouse-Geisser |
| TBW3 | 276.277 | 1.535 | 179.895 | 3.161 | 0.059 | Greenhouse-Geisser |
| Body Fat 1 | 120.824 | 1.535 | 78.687 | 1.383 | 0.253 | Greenhouse-Geisser |
| Body Fat 2 | 170.192 | 1.535 | 110.839 | 1.947 | 0.158 | Greenhouse-Geisser |
| Body Fat 3 | 48.800 | 1.535 | 31.782 | 0.558 | 0.529 | Greenhouse-Geisser |
| Upper Mid-Arm Circumference1 | 23.097 | 1.535 | 15.042 | 0.264 | 0.709 | Greenhouse-Geisser |
| Upper Mid-Arm Circumference2 | 77.877 | 1.535 | 50.718 | 0.891 | 0.389 | Greenhouse-Geisser |
| Upper Mid-Arm Circumference3 | 114.678 | 1.535 | 74.685 | 1.312 | 0.269 | Greenhouse-Geisser |
| HEI-2015 Score1 | 159.058 | 1.535 | 103.588 | 1.820 | 0.175 | Greenhouse-Geisser |
| HEI-2015 Score2 | 155.551 | 1.535 | 103.257 | 1.814 | 0.176 | Greenhouse-Geisser |
| HEI-2015 Score3 | 22.213 | 1.535 | 14.466 | 0.254 | 0.716 | Greenhouse-Geisser |

educational interventions may be valuable in promoting healthier lifestyle choices among university students, although further research is needed.

Since the first proposition of DOHaD hypothesis, the accumulated outcomes of several laboratory and clinical research point out the necessity of translating current evidence into practice but few studies suggested that women of reproductive age were not aware of the importance of early life nutrition and impact of maternal environment on growth and

development of the baby adequately. A large-scaled study undertaken in five European countries with 2,071 first-time mothers reported that diet as an infant was considered to be a less important influence on lifelong health than many lifestyle, behavioral, and environmental factors and genetics (*Gage et al., 2011*). When the evidence of the link between maternal diet quality and diet quality of the offspring at the age of 14 is considered (*Bjerregaard et al., 2019*), the wide perspective of DOHaD knowledge appears to have a great potential of preventing unhealthy lifestyle factors. Recent studies emphasize the necessity of focusing on prevention rather than treatment to break the cycle of NCDs (*Tohi et al., 2022*). One study conducted by *McKerracher et al. (2020)* developed and applied the DOHaD KNOWLEDGE scale in pregnant women and reported that diet quality during pregnancy was positively associated with DOHaD KNOWLEDGE after adjusting for sociodemographic factors. Although the DOHaD KNOWLEDGE scale introduced a set of data regarding the knowledge level related with DOHaD concepts, it consisted of five positively phrased items and targeted at pregnant women. The internal reliability of DOHaD KNOWLEDGE scale was found be satisfactorily high and for this reason authors concluded that a coherent mental construct was obtained. The need and importance of measuring and constituting baseline awareness of DOHaD concepts through systematic tools is reported, previously (*Bay et al., 2017*; *Oyamada et al., 2018*; *McKerracher et al., 2020*). The DOHaD hypothesis is based on several mechanisms including factors associated with epigenetics, inadequate and unbalanced nutrition, paternal health and mother's well-being at different stages of the reproductive period such as pre-pregnancy, pregnancy and lactation. Therefore, in the DOHaD awareness scale we aimed to establish a comprehensive list of items which comprise a wide perspective of DOHaD concepts to initiate or determine an awareness level. The study successfully produced a reliable and internally consistent tool for the defined purposes.

DOHaD education exerted positive effects on the perception of the statements "The food I eat now (11–14 years of age) will affect my health in the future" and "The food a woman eats when she is pregnant affects the health of her baby when it is grown up" as the percentage of the "strongly agree" responses increased in a study conducted on adolescents (*Bay et al., 2012*). Another study conducted by *Oyamada et al. (2018)* reported that the awareness levels of the students about the concepts of "DOHaD concept" and "First 1,000 days" increased significantly after DOHaD training in college students. Similarly, the initial scores of DOHaD awareness scale revealed a modest understanding of the DOHaD concepts since the average baseline score was 69 in the current study. The three items belonged to the questions 13, 14 and 20 ("How much does a pregnant woman's nutrition during pregnancy affect the health of the fetus in adulthood?", "How much does a pregnant woman's nutrition during pregnancy affect the health of the fetus in elderly?", and "How much can a pregnant woman's nutritional status during pregnancy affect the health of the next generations by changing gene functioning?", respectively) had the lowest scores before the education module but at the end of 6 months the scores belonged to these items revealed a greater understanding. These questions are composed of the core understanding of DOHaD concepts. Especially question 20, which interrogated the knowledge of epigenetics, introduced one of the most significant component of DOHaD

awareness scale but this item had the lowest score at 6 months after the treatment. It is known that epigenetics is currently one of the most attractive field of developmental biology and the methods used to link environmental factors and development of chronic diseases through epigenetic mechanisms will be accessible in near future (*Lynch et al., 2022*). The importance of appealing public interest to the understanding of epigenetics was specified, previously (*Lynch et al., 2022*). Therefore, we believe that tools like DOHaD awareness scale can help to take attention of public to the topic of epigenetics and its fundamentals.

Educational approaches are among the most effective methods for implementing preventive initiatives aimed at enhancing individuals' knowledge of DOHaD (*Tohi, Tu'akoi & Vickers, 2023*). Nutrition education programs and behavior change interventions can improve the knowledge and implementation of healthy eating practices (*Mahumud et al., 2022*). In this context, HEI has been considered a useful tool to assess the effectiveness of nutrition intervention programs in different studies. Expanded Food and Nutrition Education Program participation resulted in improved outcomes of HEI scores in a study involving on 97,552 participants (*Gills et al., 2021*). A consumer education programme increased HEI component scores for total fruit, whole fruit, whole grains, saturated fat, and energy from solid fats, alcohol, and added sugars significantly (*Glanz et al., 2012*). The influence of DOHaD education modules on HEI parameters is not well-known since previous studies evaluated the link between diet quality and DOHaD education used either subjective statements on eating behaviors (*Bay et al., 2012*, *2017*) or different tools. *Bay et al. (2012)* suggested the use of more substantial diet quality indicators to examine the influence of intervention in detail. *McKerracher et al. (2020)* reported that maternal diet quality which was calculated with Prime Screen©Food Frequency Questionnaire was positively associated with DOHaDKNOWLEDGE. They concluded that higher familiarity with DOHaD concepts resulted in greater diet quality during pregnancy. In the current study, HEI-2015 was used due to its reliable and generalisable properties. HEI-2015 scores of the participants significantly improved in the current study at 6 weeks and 6 months after the DOHaD education module. This effect can be explained through the motivational effect of DOHaD knowledge on positive behavioral changes. These findings highlight the potential importance of DOHaD awareness in promoting healthy lifestyle habits.

The effect of nutrition education programs and behavior change interventions on anthropometric measurements and physical activity levels has also been reported (*Barrett et al., 2021*). To date, none of the studies have evaluated the effect of DOHaD knowledge on anthropometric measurements and physical activity levels. In line with HEI-2015 scores, the participants in the current study exhibited healthier anthropometric measurements and physical activity levels at 6 months after the education module in comparison to pre-intervention period. These results indicate a coherent influence of DOHaD education on behavior change towards a healthier lifestyle. Although there are studies in the literature in which significant weight loss is achieved with single education session, this situation should not be considered a decisive point for maintaining the ideal body weight in long term (*Bhurosy & Jeewon, 2013*; *Wadolowska et al., 2019*). Therefore, it

is very crucial to provide healthy nutrition education to the public at regular intervals and to ensure their sustainability. At this point, the implementation of DOHaD concepts into practical advice with measurable goals is critically important.

The results obtained from the GEE analysis indicate that both the DoHaD awareness scale and HEI-2015 scores have a substantial impact on the anthropometric outcomes over time, with their interaction being critical for changes in the dependent variables. The AR (1) correlation structure indicates a diminishing correlation between repeated measurements over time, which is characteristic of longitudinal data. The multivariate regression analysis reveals that the waist-to-height ratio plays a significant role in influencing awareness of the DOHaD, emphasizing the relevance of abdominal fat distribution. This finding supports the notion that health risks associated with abdominal obesity increase awareness. Other anthropometric measurements and Healthy Eating Index (HEI) scores did not show a significant relationship with DOHaD awareness, and time did not have a notable effect, implying that awareness tends to remain stable. In summary, specific indicators such as the waist-to-height ratio affect health awareness, while other factors illustrate the complexity surrounding this issue.

The current study has several limitations that should be considered for future research involving the DOHaD awareness scale. First, the sample in this study consists solely of first-year nursing students, who may have a heightened concern for maintaining a healthy lifestyle. The sustained improvement in dietary quality and anthropometric measurements observed six months after the educational module may not solely reflect the impact of DOHaD concepts, but could also be influenced by the participants' overall attentiveness to health. Conducting the study without a control group limits the attribution of the findings solely to the educational module, independent of external factors, which may restrict the generalizability of the results. Therefore, there is a clear need for additional studies that explore the effectiveness of the DOHaD awareness scale within the broader population.

Future research should include a control group that does not receive the DOHaD educational module or any other intervention. This would allow for comparisons with the intervention group and help isolate the effects of the educational intervention from other external factors.

Additionally, to validate the robustness and generalizability of the results, studies should be conducted with participants from multiple locations or schools, including different genders, various academic disciplines, and different age groups.

## CONCLUSIONS

Education of DOHaD concepts through an undergraduate programme or a similar intense programme can be easily achieved due to susceptible nature of the participants. Yet, creating available conditions to translate fundamental DOHaD concepts for general public with health care workers, policymakers and researchers is urgently needed for the health of future generations. Knowledge about DOHaD, counselling on DOHaD in practice settings and impact of DOHaD on health have been identified as the three themes that should be considered in translating DOHaD in practice.

## ACKNOWLEDGEMENTS

The authors gratefully acknowledge the statistical support of Merve Kasikci from the Department of Biostatistics, Hacettepe University, and Funda Isık from the Department of Nutrition and Dietetics, Kastamonu University. Additionally, the authors would like to thank the study participants for their cooperation. During the preparation of this manuscript, OpenAI's ChatGPT, a large language model, was utilized for language editing support.

### Funding

The authors received no funding for this work.

### Competing Interests

The authors declare that they have no competing interests.

### Author Contributions

- Kadriye Elif İmre conceived and designed the experiments, performed the experiments, analyzed the data, prepared figures and/or tables, authored or reviewed drafts of the article, and approved the final draft.
- Aslı Akyol conceived and designed the experiments, performed the experiments, analyzed the data, prepared figures and/or tables, authored or reviewed drafts of the article, and approved the final draft.

### Data Availability

The raw measurements are in the Supplemental Files.

### Supplemental Information

Supplemental information for this article can be found online at http://dx.doi.org/10.7717/peerj.18669#supplemental-information.

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
