# Peer review of "Developing the developmental origins of health and disease (DOHaD) awareness scale to assess an education module for improving dietary behavior among college students"

_PeerJ, doi:10.7717/peerj.18669_

## Round 0.1 · original submission · Major Revisions

Thank you for submitting your paper to PeerJ. I have received reviews from 4 reviewers, who are all broadly in agreement regarding the contribution of your paper, but who all have suggestions for improvement. The reviews are appended below, so I will not reiterate them, but please ensure that you address all the reviewers’ comments.

Additionally, I had some comments from my own reading of the manuscript that I believe would substantially strengthen the manuscript. Although the reviewers have all indicated minor revisions, given the number of revisions (along with my own suggestions, see below), I believe that a major revision decision is more aligned with the reviews (when considered together – and the suggested restructure that I am proposing below).

Specifically, the title of the manuscript is focussed very much on development and validation of the scale, and given this, I initially thought this was going to be a psychometric paper. However, it incorporates some scale development work (which is not very clearly reported) and an intervention pre/posy study as well (and the results section actually focuses on the intervention). In addition, many of the reviewers had concerns about the intervention study, and particularly the lack of a control group.

I think these concerns could all be addressed by specifically focusing on the scale development and evaluation (i.e., explicitly making the paper a psychometric paper). This could be easily done by restructuring the paper so that that there are 4 phases, each with their own section in the results. Three of these sections are already included but are lost in large chunks of text in the methods (rather than results) and are not clearly articulated in the aims section of the introduction.

1) Phase 1: Item development – with the results section including initial scores for the item set and more information on how the final 20 items were decided (including any information on consistency between the 5 raters on the final items to be included).

2) Phase 2: Initial psychometric evaluation – with results section including the EFA and initial internal consistencies from the initial dataset. In this section I would like to see a bit more statistical information, including information on choice of rotation (and justification for this – I assume you expect factors to be correlated, so would be using an oblique rotation?) and correlations between factors.

3) Phase 3: Validation/confirmation of scale structure [this is a new analysis that I would recommend including, that will substantially strengthen the paper] – the results section should include a CFA of the scale (using the baseline data from the original “confirmatory study” – i.e., the data prior to the education intervention). Ideally, the CFA will provide additional evidence for the same 2-factor structure obtained in Phase 2 (but using confirmatory rather than exploratory analyses). By including an exploratory and confirmatory analysis in the same paper, I think this would substantially improve the evidence for the new scale and the contribution of the paper to the literature.

4) Phase 4: Sensitivity to change. Rather than including the education program as an intervention, you could use this psychometrically to look at new scales sensitivity to change – i.e., where there any changes in the two subscales pre-post taking part in the intervention? By focusing on measurement and sensitivity, the lack of a control group is less critical, and would mean the anthropometric measures are no longer critical to the primary aim (i.e., sensitivity to change) but are simply included as validation measures that the education program was associated with some change (and the anthropomorphic measures could even be shifted to supplementary materials if word count is an issue with the additional CFA included in Phase 3).

If you had not included the intervention, then I would suggest that this phase is also where you could also report on test-retest reliability (and simply reported correlations/ICC between scores at the two time points). But, given you include an intervention, this a problem – as this is probably in direct competition with sensitivity to change. So, I would suggest focussing on sensitivity to change, and establishing the measure can be used to assess change in the context of intervention. If the scale is not sensitive to change (which may be the case given high ICCs), then this should also be reported (and the test-retest reliability used to indicate that maybe it is a better trait than state measure).

Thank you again for submitting your paper to PeerJ, and I hope you find these comments and those of the reviewers useful in revising your manuscript.

Reviewer 1 ·

Basic reporting

Since it is possible and likely that the DOHaD hypothesis is incorrect and the findings that support the hypothesis are due to the statistics used (over-adjustment) in the pro-DOHaD studies, this study’s introduction need to describe this possibility. The paper should cite several key manuscripts that described this problem:
a. Kramer MS, Zhang X, Dahhou M, Yang S, Martin RM, Oken E, et al. Does fetal growth restriction cause later obesity? Pitfalls in analyzing causal mediators as confounders. Am J Epidemiol. 2017 Apr;185(7):585–90.
b. Huxley R, Neil A, Collins R. Unravelling the fetal origins hypothesis: is there really an inverse association between birthweight and subsequent blood pressure? Lancet 2002 Aug;360(9334):659–65.
c. Ananth C V., Schisterman EF. Confounding, causality, and confusion: the role of intermediate variables in interpreting observational studies in obstetrics. Am J Obstet Gynecol [Internet]. 2017;217(2):167–75.
d. Schisterman EF, Cole SR, Platt RW. Overadjustment bias and unnecessary adjustment in epidemiologic studies. Epidemiology [Internet]. 2009;20(4):488–95.
e. Paneth N, Ahmed F, Stein AD. Early nutritional origins of hypertension: a hypothesis still lacking support. J Hypertens Suppl. 1996 Dec;14(5):S121–9.
f. Elmrayed S, Metcalfe A, Brenner D. Wollny K. & Fenton TR. Are small-for-gestational-age preterm infants at increased risk of overweight? Statistical pitfalls in overadjusting for body size measures. J Perinatol (2021). PMID: 33850286 https://rdcu.be/ciB7k
g. Williams TC, Bach CC, Matthiesen NB, Henriksen TB, Gagliardi L. Directed acyclic graphs: a tool for causal studies in paediatrics. Pediatr Res. 2018 Oct;84(4):487-493. PMID: 29967527; PMCID: PMC6215481.
h. Jain S, Samycia L, Elmrayed S, Fenton TR. Does the evidence support in utero influences on later health and disease? A systematic review of highly cited Barker studies on developmental origins. J Perinatol. 2024 Feb 9. doi: 10.1038/s41372-024-01889-4. Epub ahead of print. PMID: 38337020.
Given that the DOHaD hypothesis might be incorrect, it would be preferable to omit “substantially” from the abstract.

Experimental design

In the abstract “can help to support combating with the development of chronic diseases” should be changed to “can help to support improved lifestyle behaviors” since the students were not followed to assess their development of chronic diseases.

Validity of the findings

The abstract is not clear about whether measurements were made once as the methods say: “pre-intervention period in which the baseline parameters” or twice. The results suggest twice “resulted in improved…”.

Reviewer 2 ·

Basic reporting

no comment

Experimental design

Although the study design is generally reasonable and systematic, there are several areas where improvements could be made to enhance its scientific rigor and the credibility of the results.

1. Lack of a Control Group: The study design does not mention the inclusion of a control group that does not receive the educational module, making it difficult to establish a causal relationship between the intervention and the observed changes. Future studies should incorporate a control group that does not receive the DOHaD educational module. This would allow for comparisons with the intervention group, helping to isolate the effects of the educational intervention from other external factors.

2. Sample Diversity: The study sample is limited to first-year female nursing students, which may restrict the external validity of the results. Future research should include participants of different genders, various academic disciplines, and different age groups. Increasing sample diversity would. Conducting the study at multiple locations or schools could further validate the robustness and generalizability of the results.

3. Multivariate Analysis: The current analysis primarily focuses on univariate and bivariate analysis, without delving into the interactions between multiple variables. Introduce multivariate regression analysis or structural equation modeling (SEM) to explore complex relationships between different variables and potential moderating and mediating effects.

4. Handling Repeated Measures: The article mentions using repeated measures ANOVA to handle quantitative variables but does not detail how it addresses autocorrelation in repeated measures. Utilize mixed-effects models or generalized estimating equations (GEE) to better handle autocorrelation and variability issues in repeated measures data.

Validity of the findings

Report of the results: Use Equator metwork guidelines and checklist to help you report your results. Such as Guidelines for reporting reliability and agreement studies (GRRAS).

·

Basic reporting

Clear language was used throughout with the occasional mistake. Examples are occasionally omitting "the" and the specific examples listed here:
1. Please be consistent with spelling of “fetus” in the manuscript
2, Please be consistent with spelling of “DOHaD” in the manuscript
3. Please note that “Likert scale” has a capital L.
The background and references were sufficient, the structure and figures were excellent except for these specific examples:
4. I can’t see any reference to the supplementary data; please add.
5. Line 266 & Table 3: please add individual P-values for 6 weeks after intervention for comparison
6. Line 291-312: please consider a (supplementary) table or graph to illustrate your findings here
Other suggested corrections are:
7. Line 33: Please explain what HEI-2015 is in the abstract
8. Lines 56-57 “Preventable environmental factors such as smoking, sedentary life style, unbalanced and inadequate nutrition are the major causes of NCDs”. Would war, famine and poverty also contribute significantly as they are to some extent preventable?
9. Lines 342-344: please correct the formatting here
10. I can’t see any reference to the supplementary data; please add.
Otherwise, the manuscript was self-contained with relevant results to the main aim, although no hypothesis was provided.

Experimental design

My main criticism is that the authors should have had controls with no intervention because students could have picked up similar concepts from their nursing course. I would at least like them to discuss this limitation.
The main aim and research question is clear: to develop and validate DOHaD Awareness Scale, and examine whether having an DOHaD education module may affect dietary behaviour in college students.
The study definitely fills a knowledge gap about how DOHaD knowledge can lead to self-directed health changes.
Methods were clear and self-contained with minor issues to address:
11. Lines 155-6: please elaborate on what these two tests do and what their scales are.
12. Line 177: please add a reference for the Healthy Eating Index (HEI-2015)
13. Line 181: please insert “The” before ”education programme”
14. Line 193: were anthropometric measurements measured once or in triplicate and averaged?

Validity of the findings

Although similar studies have been performed, this study's findings are novel and valid in themselves and a worthy addition to the literature.
All underlying data was provided and the analysis was robust and statistically sound. Please also see my criticism of controls in Experimental Design.
Conclusions are generally well stated and linked to the original research question.

Additional comments

No comment

Reviewer 4 ·

Basic reporting

The manuscript “Development and preliminary validation of the developmental origins of health and disease (DOHaD) awareness scale” addresses the importance of an awareness scale related to the DOHAD concept, and how an awarreness scale can aid kwnoledge of DOHAD concept and how it can be used as a preventative way to avoid NCDs.

The manuscript is clear, well written, the authors use a technically correct text. The authors describe very well the problem about the lack of knowledge of the DOHAD concept and the relevance and impact it has on later life, as well as the importance of maternal diet and nutritional status.

However, I had some comments, related majority on to the objective of the manuscript. For example,

In lines. 43 to 47, “In order to achieve this goal, systematic scales like DOHaD Awareness Scale may aid to create individual and social consciousness to prevent and protect against unhealthy nutrition and related chronic diseases for future generations”.

But then in line 107 to 110 the authors declarate. “In this study, DOHaD Awareness Scale, which consists of twenty items that lists some important aspects of DOHaD concepts, including epigenetics, social interactions, and life stage from pre-pregnancy to the age of two, was developed primarily to measure the awareness of healthcare professionals who are in the education stage.

It´s described that the DOHaD Awareness Scale, was developed primarily to measure the awareness
of healthcare professionals who are in the education stage.

There is no discussion about how the educational module was designed. Neither about the importance of the educational module as an intervention form, because although the scale helps to create awareness with just the questions, if people do not have an intervention with an educational module, probably they would not change their eating habits, because if they do not know the repercussions of bad habits at the level of the DOHAD concept, this study could not conclude that systematic scales like DOHaD Awareness Scale may help to create individual and social consciousness.

The concept could be treated as the authors describe it in the line 99 ..

....defining the baseline awareness and understanding of DOHaD concepts...

Both concepts together.

Or as the authors describe in line 103, and highlight why other authors used basic questionnaires rather than a systematically developed and validated scale, and also highlight how or why their DoHAD scale awarenees can be used, and the relevance of their manuscript.

Its no described if the validated qustionare and the scale are relevant for the type of intervention or the type of information that the persons has to received about dohad concepts


Line 108. The epigenetic concept has not been described previously, so it is necessary to describe it in order to highlight its importance.

Since one of the main objectives of the manuscript is the creation and validation of the sensitization scale, the possible answers to the questionnaire, as well as a more detailed description of the results obtained from the questionnaires, should be included as results in a table in the main manuscript.


Table 1 is not displayed in full.

A deep analysis of DOHAD awareness questionnaire data obtained should be considered. The results of anthropometric measurements are very important, but, since the title of the article is precisely about awareness scale, more data about it can be added. Also, for example a correlation could be made between the increase in exercise and the decrease in body weight, after the DOHAD awareness questionnaire was applied. The results of the measurements would be more solid.


Relate to the discussion section.

Lack of discussion about the possible explanation of why anthropometric measurements decreased or why the average physical activity of light and moderate exercise increased after the application of the questionnaire. As well as why vigorous exercise did not change after dohad knowledge.

More new recent references related to the topic should be added, for example the last papers of Vickers MH. PMID: 27417627; PMID: 38258455; PMID: 35627561

Experimental design

The manuscript describe original research within the Aims & Scope of the Journal.

The manuscript doest not clearly define the research question, which must be relevant and meaningful. The knowledge gap being investigated should be identified, and statements should be made as to how the study contributes to filling that gap.

Methods are described with sufficient information to be reproducible by another investigator.

The research must have been conducted in conformity with the prevailing ethical standards in the field.

Validity of the findings

no commnet

---

## Round 0.2 · Minor Revisions

Thank you for submitting your revised manuscript to PeerJ. I have now received reviews from four reviewers and believe that if the minor comments of Reviewers 1 and 3 can be addressed, then the paper will likely be acceptable for publication in PeerJ. In particular, the use of causal language needs to be removed throughout the manuscript given the lack of a control group (see comments from Reviewer 1). In addition, given my concerns over the psychometric analyses, I agree with Reviewer 1 that the language and conclusions throughout the paper (but in particular the abstract) need to be tempered (e.g., "may be", "could" etc).

While, my concerns regarding the psychometric analyses remain, given these were not echoed by the reviewers, I believe these can be addressed by simply minimising the focus on psychometric information in the title of the paper (to avoid any confusion regarding psychometric quality). An additional advantage of this is that it would also allow the intervention component of the manuscript to be reflected in the title (which at the moment is totally absent). Therefore, I suggest changing the title to: "Developing the developmental origins of health and disease (DOHaD) awareness scale to assess an education module for improving dietary behaviour among college students".

Reviewer 1 ·

Basic reporting

Plenty of over statements of evidence

Experimental design

The text and abstract both need to acknowledge “Despite the lack of control group” and therefore should not suggest cause and effect language (resulted in) but rather refer to “associated with”.

Validity of the findings

The text and abstract both need to acknowledge “Despite the lack of control group” and therefore should not suggest cause and effect language (resulted in) but rather refer to “associated with”.

Additional comments

1.Given that the DOHaD hypothesis might be incorrect, it is important to omit “substantially” from the abstract. Despite your reponse that this wording was changed, this error has NOT been corrected. That abstract sentence should be changed to: Some studies conducted within the scope of Developmental Origins of Health and Disease (DOHaD) hypothesis show associations between early-life environmental factors, especially maternal health status and nutritional status, with next generation’s health and disease status. xxx

2. The abstract conclusion should be changed to: “The use of the DOHaD Awareness Scale may be useful to encourage healthier behaviours among college students.” It should not mention chronic diseases or future generations since these outcomes were not measured in this study.

Reviewer 2 ·

Basic reporting

The article's basic reporting meets the journal’s standards, with clear language, a thorough background, appropriate structure, and well-prepared figures and tables.

Experimental design

The design of this study is well-structured, with clear phases of scale development, validation, and application. However, the study focuses solely on first-year nursing students, which may limit the generalizability of the findings. Future research could benefit from including more diverse participant groups. And the lack of a control group in the confirmatory phase reduces the ability to isolate the effect of the educational intervention from other external factors.

Validity of the findings

The article's findings are presented as valid due to comprehensive data transparency, sound statistical approaches, and supported conclusions, though the authors recommend further validation in diverse populations for generalizability​.

·

Basic reporting

Just minor points:
1. Ensure consistency in the use of British and American English. For example, "behaviour" vs. "behavior".
2. Check for consistent use of punctuation, especially commas and periods.
3. Ensure all references are correctly formatted and consistent.

Experimental design

All OK

Validity of the findings

Oll OK

Additional comments

I am satisfied that the authors addressed by feedback satisfactorily. I have 3 minor corrections:
1. "Non-communicable diseases (NCDs), including cardiovascular heart diseases, diabetes, stroke and cancer, are the leading cause of mortality and morbidity in our century (Wang &Wang, 2020)." should be "Non-communicable diseases (NCDs), including cardiovascular diseases, diabetes, stroke, and cancer, are the leading causes of mortality and morbidity in our century (Wang & Wang, 2020)" (Line 23).
2. "The study consisted of three phases: identification of DOHaD awareness scale components, development, validation, interrater reliability of the scale, and confirmatory study." should be "The study consisted of three phases: identification of DOHaD awareness scale components, development and validation, interrater reliability of the scale, and a confirmatory study" (Line 17).
3. "Translating DOHaD concepts to healthier behaviours can help to support improved lifestyle behaviors." should be "Translating DOHaD concepts to healthier behaviors can help support improved lifestyle behaviors" (Line 20).

Reviewer 4 ·

Basic reporting

No comments

Experimental design

No comments

Validity of the findings

No comments

Additional comments

My recognition to the authors for the responses to the comments on the manuscript, "Development and preliminary validation of the developmental origins of health and disease (DOHaD) awareness scale", they were very clear and all were answered appropriately.

---

## Round 0.3 · accepted · Accept

Thank you for addressing the reviewers comments and editing the title to reduce the emphasis on psychometrics.